# Biomechanical Characterisation of Thoracic Ascending Aorta with Preserved Pre-Stresses

**DOI:** 10.3390/bioengineering10070846

**Published:** 2023-07-17

**Authors:** Shaiv Parikh, Kevin M. Moerman, Mitch J. F. G. Ramaekers, Simon Schalla, Elham Bidar, Tammo Delhaas, Koen Reesink, Wouter Huberts

**Affiliations:** 1Department of Biomedical Engineering, CARIM School for Cardiovascular Diseases, Maastricht University, 6229 ER Maastricht, The Netherlands; s.parikh@maastrichtuniversity.nl (S.P.); tammo.delhaas@maastrichtuniversity.nl (T.D.); k.reesink@maastrichtuniversity.nl (K.R.); 2Department of Mechanical Engineering, University of Galway, H91 TK33 Galway, Ireland; kevin.moerman@nuigalway.ie; 3Department of Cardiology, Heart & Vascular Centre, Maastricht University Medical Centre, 6229 HX Maastricht, The Netherlands; mitch.ramaekers@mumc.nl (M.J.F.G.R.); s.schalla@mumc.nl (S.S.); 4Department of Radiology and Nuclear Medicine, Maastricht University Medical Centre, 6229 HX Maastricht, The Netherlands; 5Department of Cardiothoracic Surgery, Heart & Vascular Centre, Maastricht University Medical Centre, 6229 HX Maastricht, The Netherlands; elham.bidar@mumc.nl; 6Department of Biomedical Engineering, Cardiovascular Biomechanics, Eindhoven University of Technology, 5612 AE Eindhoven, The Netherlands

**Keywords:** inverse finite element analysis, in vivo zero pressure geometry, pre-stressing algorithm

## Abstract

Mechanical properties of an aneurysmatic thoracic aorta are potential markers of future growth and remodelling and can help to estimate the risk of rupture. Aortic geometries obtained from routine medical imaging do not display wall stress distribution and mechanical properties. Mechanical properties for a given vessel may be determined from medical images at different physiological pressures using inverse finite element analysis. However, without considering pre-stresses, the estimation of mechanical properties will lack accuracy. In the present paper, we propose and evaluate a mechanical parameter identification technique, which recovers pre-stresses by determining the zero-pressure configuration of the aortic geometry. We first validated the method on a cylindrical geometry and subsequently applied it to a realistic aortic geometry. The verification of the assessed parameters was performed using synthetically generated reference data for both geometries. The method was able to estimate the true mechanical properties with an accuracy ranging from 98% to 99%.

## 1. Introduction

The functionality of the aorta to act as a conduit for blood from the heart to the systemic circulation is supported by its microstructure [1]. The microstructure mainly comprises elastin and collagen as major load-bearing constituents, which impart elasticity and strength, respectively, to the aortic wall [2]. Variations in concentrations, reorientation, or change in mechanical properties of these constituents reflect changes in the biomechanics of the aorta at a structural level [1,2,3]. For example, it has been shown that a stiffening of the aortic wall by the loss of elastin integrity is associated with an enlargement of the aortic aneurysms [2,4]. Therefore, the estimation of aortic mechanical properties might prove crucial for understanding the pathological changes associated with the tissue.

The use of inverse finite element analysis (IFEA) is a common approach in biomechanics for a non-invasive assessment of mechanical properties [5]. IFEA typically involves creating a finite element (FE) geometry of the object of interest, assuming a constitutive law with initial mechanical property parameters, applying realistic loads to the FE geometry and computing the outcome in terms of motion and deformation. The computed outcomes are then compared to in vivo experimental data, such as measured deformations. The mechanical parameters are iteratively modified until best-fit estimates are obtained [5]. However, to obtain accurate mechanical parameters using IFEA, an unloaded (FE) geometry of the object of interest is essential since all the realistic loads are applied on the unloaded geometry.

Aortic geometries can be obtained through the segmentation of medical image data obtained from computed tomography (CT), magnetic resonance imaging (MRI) or ultrasound [6,7]. Although stressed in vivo due to a pre-existing strain and a stress field in the vessel wall, the aortic geometries created from the medical images are not stressed [6,7]. Considering the obtained geometries as zero-stress configurations will result in incorrect values of stresses in the vessel wall during IFEA, and consequently an incorrect parameter estimation for non-linear elastic materials [6,7]. Therefore, the determination of the pre-stresses remain imperative to obtain physiologically relevant mechanical parameters during IFEA [6,7].

Prior research has shown that it is possible to achieve a realistic stress state in vivo by using algorithms that estimate pre-stresses from pressurising an unloaded configuration [6,7,8,9,10,11,12], or by considering the stress field to be statically determinate [13,14]. However, most of the algorithms mentioned in the studies [6,7,8,9,10,11,12] require prior knowledge of mechanical parameters obtained through experimental studies on tissue specimens in order to estimate the unloaded configuration, whereas the statically determinate consideration of stress field in [13] led to inaccurate wall stress estimations. The backward incremental (BI) method [7,11] has demonstrated an effective estimation of mechanical parameters and pre-stresses. It applies pressure incrementally to build up pre-stresses in the vessel wall while discarding the related deformations. This involves updating the initial stress field in each element with each pressure increment, making the process of building pre-stresses in vessel walls challenging using the BI method [6]. Essentially, without experimentally determined mechanical parameters, existing pre-stressing algorithms cannot estimate the unloaded configuration. Furthermore, the task of estimating mechanical parameters through BI is arduous because it requires modifying FE structural solvers.

In this paper, we propose an IFEA algorithm that can identify mechanical properties non-invasively while considering pre-stresses. Our algorithm eliminates the need for updating the initial stress field and combines fixed-point iterative schemes from previous studies [6,10] to determine zero-pressure geometries. For validation, we applied our method to the synthetically generated reference data of a simple cylindrical geometry first and subsequently to that of a complex, real aortic geometry.

## 2. Materials and Methods

The focus of this paper is to validate a mechanical parameter estimation algorithm for a general aortic geometry. The validation of the algorithm is performed by implementing the workflow on synthetically derived reference data as a substitute for patient-derived experimental data. Therefore, to understand the validation process, the algorithm must be read by replacing “experimental data” with “synthetic reference data”.

The key elements required for the mechanical parameter estimation entail geometry creation (in this paper: cylindrical geometry and an image-based aortic geometry), an FE model and an inverse modelling algorithm that considers existing pre-stresses. Each of these components are described in this section. At the end of the section, the creation of synthetic reference data and the simulations that were performed to assess the accuracy of our IFEA are presented. 

### 2.1. Vessel Geometries

In this section, the geometry creation of our two computational domains, i.e., cylinder and aorta, will be explained. Additionally, the boundaries where pressures and displacements are applied are also explained. All operations of geometry creation were performed using MATLAB codes developed based on the open source GIBBON toolbox [15].

#### 2.1.1. Cylindrical Geometry

A cylindrical geometry was created with a radius of 16 mm and a length of 192 mm. Subsequently, the geometry was meshed using quadrilateral elements. The mesh was thickened by 1 mm on the inner side to create an 8-noded hexahedral element mesh. This resulted in an inner radius of 15 mm for the entire geometry. Following a mesh convergence analysis, in which a uniform pressure was applied to the inner surface of the geometry, the mesh density was gradually increased until the relative change in the predicted mean von Mises Cauchy stress was less than 1%. As a result, the geometry consisted of 2176 hexahedral elements and 4420 nodes (denoted as $X_{cylinder}^{unload}$). The nodes on the openings of the cylinder ($\Gamma_{d}$; cf. Figure 1) were provided with fixed boundary conditions during all simulations, while the inner surface of the cylinder ($\Gamma_{n}$; cf. Figure 1) was prescribed with pressure load.

#### 2.1.2. Image-Based Aorta Geometry

The aortic geometry at diastolic pressure was reconstructed from DICOM data of the 3D DIXON sequence generated from 3-T MR systems (Philips Ingenia; Philips Healthcare, Best, The Netherlands). From the DICOM data, x,y and z coordinates of four points around the luminal edges were retrieved for six anatomical locations (Figure 2A). 

Subsequently, smooth ellipses were fitted through the retrieved four points from the luminal edges. The outer boundaries were then created by adding a layer of points at an offset of 2 mm from the inner boundaries (at each plane location). A curve connecting the centroids of the outer-wall boundaries through a cubic polynomial fit represented the centre-line of the main trunk of the aorta. Similar to the cylindrical model, a quadrilateral surface-mesh was swept along the centre-line of the main trunk. Next, major branch vessels were added. The mesh was thickened in the inwards direction (2 mm thickness) to create 8-noded hexahedral elements (Figure 2B). Based on mesh convergence study (for procedure, cf. 2.1.1), the geometry resulted in 11,998 elements with 24,236 nodes (denoted as $X^{MRI}$). All boundary conditions were applied on the unloaded configuration (detailed explanation in next section). The nodes on openings ($\Gamma_{d1}$ to $\Gamma_{d4}$; Figure 2B) were prescribed with fixed (Dirichlet) boundary conditions while pressurising the inner surface $\Gamma_{n}$ (Figure 2B) of the unloaded configuration to diastolic pressure. However, the nodes at the root level $\Gamma_{root}$ (Figure 3B) were prescribed with displacement while pressurising the unloaded configuration to systolic pressure to take into account the longitudinal motion of the root during a cardiac cycle. The root movement was determined as follows: the aortic valve movement was derived from single slice cine images of the left ventricular outflow tract (LVOT), which was acquired in two directions (perpendicular to each other). Valve displacement was measured automatically by using commercially available software (CAAS MR Solutions v 5.2.1, PieMedical Imaging, Maastricht, the Netherlands).

### 2.2. Finite Element Analysis

The finite element model is governed by the balance of linear momentum applied to the computational domain Ω (either the cylinder or aorta) and a constitutive model of the vessel wall.

The balance of linear momentum in the absence of body forces, and after neglecting inertial effects, reads:(1)$$\nabla \cdot \overset{̿}{\sigma }=0$$
where, ∇ is the gradient operator and $\overset{̿}{\sigma }$ is the second-order Cauchy stress tensor. 

To render the solution of the balance of linear momentum possible, boundary conditions (Dirichlet and Neumann) need to be satisfied where applicable.

Dirichlet boundary conditions are given by:(2)$$x_{p}\cdot n_{p}=x_{p}$$
where, $x_{p}$ represents the prescribed nodal displacements in the direction $n_{p}$, on surfaces $\Gamma_{d}$ (Figure 1 and Figure 2).

Neumann boundary conditions are given by:(3)$$(\overset{̿}{\sigma }\cdot n_{a})\cdot n_{a}=s_{a}$$
where, $s_{a}$ denotes the magnitude of applied traction in the direction $n_{a}$ on surfaces $\Gamma_{n}$. As seen in Figure 1 and Figure 2, the Neumann boundary condition in this study only comprises blood pressure (p) application on the luminal side of the vessel.

An uncoupled Ogden hyperelastic strain energy density (Ψ) formulation [16] was used to describe the mechanical behaviour of the vessel geometries. The strain energy density (Ψ) was represented as:(4)$$\Psi =\Psi_{vol}+\Psi_{iso}$$
where, $\Psi_{vol}=\frac{\kappa }{2}{ln(J)}^{2}$ is the volumetric and $\Psi_{iso}=\sum_{a=1}^{N}\frac{c_{a}}{m_{a}}({\tilde{\lambda }}_{1}^{m_{a}}+{\tilde{\lambda }}_{2}^{m_{a}}+{\tilde{\lambda }}_{3}^{m_{a}}-3)$ is the isochoric part of the strain energy density function. The parameter $c_{a}$ is the shear modulus-like mechanical parameters, parameter κ is the mechanical bulk modulus, while the mechanical parameter $m_{a}$ is unit-less and controls the degree of nonlinearity. The parameter N defines the order of the Ogden model. Quantity J is the volume ratio defined by $J=\lambda_{1}\lambda_{2}\lambda_{3}$. In Equation (4), ${\tilde{\lambda }}_{i}$ are the deviatoric principle stretches, defined by ${\tilde{\lambda }}_{i}=J^{-1/3}\lambda_{i}$. Here, a second order form is used, i.e., N=2, with the following constraints: $c_{1}=c_{2}=c$, $m_{1}={-m}_{2}=m=2$, causing the model to reduce to a Mooney–Rivlin form with tension–compression symmetry [16]. To enforce nearly incompressible tissue behaviour, the bulk modulus (κ) was constrained to κ=100c. 

The mechanical equilibrium condition is said to be reached when the balance of linear momentum is satisfied (Equation (1)). The equilibrium solution resulting in the deformed shape and the stresses in the elements were obtained using FEBio’s [17] structural solver. The structural solver is denoted by S (Figure 1). By utilising the GIBBON toolbox, we automated the essential tasks of generating the FEBio model, triggering the FEBio simulation and importing the FEBio output files for further analysis in MATLAB (MATLAB R2018a; The MathWorks, Natick, MA, USA).

### 2.3. Mechanical Parameter Estimation

This section provides a general description of the implementation of the data used for fitting and the mechanical parameter estimation algorithm particularised for aortic geometries. The aortic geometries are obtained from in vivo medical images.

#### 2.3.1. Experimental Data Used for Fitting

In most cases, imaging data for the aortic geometries is obtained only for two pressure conditions, namely diastolic and systolic pressures. Henceforth, in this study, only the radii of the aortic geometry at diastolic and systolic pressures ($R_{DBP}$ and $R_{SBP}$, respectively) are used to assess the accuracy of our fitting procedure.

#### 2.3.2. Algorithm Description

The algorithm is commenced by first assigning initial guesses for the mechanical parameter c (cf. Equation (4)) and scaling parameter γ. Second, the image-extracted geometry at diastolic pressure, $\Omega \left(x_{\alpha ,dias}^{exp},0\right)$, is pressurised (subscript α denotes the geometry under consideration) to systolic pressure:(5)$$\Omega (x_{int},{\overset{̿}{\sigma }}_{int})=S(\Omega (x_{\alpha ,dias}^{exp},0),\;p_{sys})$$

Which results in the “intermediate” configuration $\Omega (x_{int},{\overset{̿}{\sigma }}_{int})$. However, note that this intermediate configuration was obtained by assuming zero pre-stresses in the image-extracted geometry and thus not the correct unloaded configuration. 

A displacement field is obtained by subtracting the nodes of the initial diastolic geometry from the intermediate configuration ($U_{\alpha }^{disp}$ = $x_{int}-x_{\alpha ,dias}^{exp}$). To consider the pre-stresses that are present in the diastolic geometry, an estimation of the unloaded configuration is made in the next step of the algorithm by subtracting a scaled displacement field from the initial diastolic configuration:(6)$$X_{\alpha }^{unload}=x_{\alpha ,dias}^{exp}-\gamma \cdot U_{\alpha }^{disp}$$

In which γ is an unknown scaling factor that needs to be estimated in addition to the mechanical parameter.

Hence, the algorithm was designed with a two-staged approach. The first stage was to estimate the value of c by assuming that the assigned γ is the correct value. The second stage comprised utilising the newly estimated c to estimate γ.

For the initially assumed value of c, the unloaded configuration is then pressurised to diastolic and systolic pressures to obtain the deformed configuration for the respective pressures (steps 1 and 2; Figure 3; Equation (7)):(7)$$\Omega \left(x_{\alpha ,dias}^{est},{\overset{̿}{\sigma }}_{dias}^{est}\right)=S(\Omega \left(X_{\alpha }^{unload},0\right),p_{dias})\;\Omega \left(x_{\alpha ,sys}^{est},{\overset{̿}{\sigma }}_{sys}^{est}\right)=S(\Omega \left(X_{\alpha }^{unload},0\right),p_{sys})$$

The inner radii of the diastolic and systolic geometries, $r_{DBP}$ and $r_{SBP}$, respectively, are calculated as the mean of distances from the luminal nodes to the centre (mean of nodal coordinates) of the lumen under consideration. The operator Φ (Figure 3: Steps 3 and 4; Equation (8)) determines the procedure to extract the inner radii.
(8)$$r_{DBP}=\Phi (S\left(\Omega \left(x_{\alpha ,dias}^{est}\right)\right))\;r_{SBP}=\Phi (S\left(\Omega \left(x_{\alpha ,sys}^{est}\right)\right))$$

The first-stage optimisation is based on minimising an objective function (Equation (9)) comprising the differences between the experimental internal radii of diastolic and systolic geometries, $R_{DBP}$ and $R_{SBP}$, and $r_{DBP}$ and $r_{SBP}$ (Step 5; Figure 3), by changing mechanical parameter c. Optimisation of the objective function leads to a new value of the mechanical parameter $c^{*}$:(9)$$c^{*}=argmin||{(R}_{SBP}-r_{SBP})-(R_{DBP}-r_{DBP})||$$

Next, the diastolic configuration $\left(x_{\alpha ,dias}^{est*}\right)$ is produced by pressurising the unloaded configuration with $c^{*}$ as the new mechanical parameter (Figure 3 Step 6):(10)$$\Omega \left(x_{\alpha ,dias}^{est*},{\overset{̿}{\sigma }}_{dias}^{est*}\right)=S(\Omega \left(X_{\alpha }^{unload},0\right),\;p_{dias})$$

The second-stage optimisation is based on minimising an objective function (Equation (11)) comprising L^2^—the norm of the nodal differences of the experimental diastolic FE geometry ($x_{\alpha ,dias}^{exp})$ and the diastolic geometry $\left(x_{\alpha ,dias}^{est*}\right)$ produced using the new mechanical parameter $c^{*}.$ The L^2^–norm is defined as $\sqrt{\sum_{i=1}^{n}{\left({\left|x_{\alpha ,dias}^{exp}-x_{\alpha ,dias}^{est*}\right|}_{i}\right)}^{2}}$ for all nodes =1,2…n; and represented as ${\left|\right|x_{\alpha ,dias}^{exp}-x_{\alpha ,dias}^{est*}\left|\right|}_{2}$. New scaling factor $\gamma^{*}$ was determined as an outcome of the second-stage optimisation (Steps 6 and 7; Figure 3):(11)$$\gamma^{*}=argmin\left|\right|x_{\alpha ,dias}^{exp}\;-\;x_{\alpha ,dias}^{est*}{\left|\right|}_{2}$$

Subsequently, a new unloaded configuration is formed using $\gamma^{*}$ (Step 8; Figure 3):(12)$$X_{\alpha }^{unload*}=x_{\alpha ,dias}^{exp}-\gamma^{*}\cdot U_{\alpha }^{disp}$$

Furthermore, the new unloaded configuration is pressurised to obtain the internal radii at diastolic and systolic pressures ($r_{DBP}^{*}$ and $r_{SBP}^{*}$; Figure 3: steps 9–10):(13)$$r_{DBP}^{*}=\Phi (S(\Omega \left(X_{\alpha }^{unload*},0\right),\;p_{dias}))\;\;\;\;\;r_{SBP}^{*}=\Phi (S(\Omega \left(X_{\alpha }^{unload*},0\right),\;p_{sys}))$$

The differences between the experimental internal radii ($R_{DBP}$ and $R_{SBP}$) and the newly obtained internal radii ($r_{DBP}^{*}$ and $r_{SBP}^{*}$) are compared by setting a tolerance value (ε) 0.01 mm. The decision to continue the iterations to estimate c and γ continues until the tolerance value is met (Figure 3). The parameters *c* and γ are optimised in a least squares sense using the MATLAB function lsqnonlin, which is configured to use the Levenberg–Marquardt algorithm [18].

### 2.4. Method Validation

The goal of validating the algorithm entailed determining the unloaded configuration and estimating the true mechanical parameter (c). Since the unloaded configuration and the true mechanical parameter cannot be obtained in vivo, reference data were generated synthetically to validate the algorithm.

#### 2.4.1. Synthetically Created Reference Data

Although reference data were generated for each geometry type, the method of generating the reference data for each geometry type differed slightly. The next two sections will delineate the methods to generate reference data ($x_{\alpha ,dias}^{exp}$, $U_{\alpha }^{disp}$, $R_{DBP}$ and $R_{SBP}$; cf. Figure 3) for validating the algorithm.

#### 2.4.2. Cylindrical Geometry Reference Data

After the creation of the unloaded geometry $\Omega \;(X_{cylinder}^{unload},0)$ (Section 2.1.1; Figure 4), it was initially pressurised to a diastolic pressure ($p_{dias}=$ 80 mmHg) to obtain the diastolic geometry $\Omega (x_{cylinder,dias}^{exp},{\overset{̿}{\sigma }}_{dias}^{exp})$ (Equation (14); Figure 4). Subsequently, the unloaded geometry was also pressurised to a systolic pressure ($p_{sys}=$130 mmHg), hereby obtaining the systolic geometry $\Omega (x_{cylinder,sys}^{exp},{\overset{̿}{\sigma }}_{sys}^{exp})$ (Equation (15); Figure 4). The transformations to diastolic and systolic geometries are shown by the following equations:(14)$$\left(x_{cylinder,dias}^{exp},{\overset{̿}{\sigma }}_{dias}^{exp}\right)=S(\Omega (X_{cylinder}^{unload},0),\;p_{dias})$$
(15)$$\Omega (x_{cylinder,sys}^{exp},{\overset{̿}{\sigma }}_{sys}^{exp})=S(\Omega (X_{cylinder}^{unload},0),\;p_{sys})$$

The displacement field $U_{cylinder}^{disp}$ was calculated to be used as an input for mechanical parameter estimation:(16)$$U_{cylinder}^{disp}=x_{cylinder,sys}^{exp}-x_{cylinder,dias}^{exp}$$

Additionally, $R_{DBP}$ and $R_{SBP}$ were created from nodes at the central inner boundary of the cylinder using:(17)$$R_{DBP}=\Phi (S\left(\Omega \left(x_{cylinder,dias}^{exp}\right)\right))\;R_{SBP}=\Phi (S\left(\Omega \left(x_{cylinder,sys}^{exp}\right)\right))$$

The reference data were generated for three different values of the shear modulus-like mechanical parameter (c; cf. Equation (4))—0.9 MPa, 1 MPa and 1.1 MPa (Figure 4). 

#### 2.4.3. Aortic Geometry Reference Data

The generation of the reference data for the aortic geometry differed slightly from the cylindrical geometry. The initial aortic geometry $\Omega (X^{MRI},0)$ (Section 2.1.2; $\Omega_{MRI}$ in Figure 5) was created from an MRI which was already at diastolic pressure, as opposed to the cylindrical geometry which was created with the unloaded configuration as the initial state. Therefore, an additional operation had to be performed on the initial aortic geometry to create an unloaded configuration (Steps 1,2 Figure 5). The initial aortic geometry was pressurised (with zero displacement at root level) with a systolic pressure ($p_{sys}=$130 mmHg) to obtain an intermediate configuration $\Omega (x_{int},{\overset{̿}{\sigma }}^{*})$ (Equation (18); $\Omega_{int}$ in Figure 5). The resulting displacement field $U_{aorta}^{disp}$ was computed (Equation (19)) and then scaled by a dimensionless scaling factor γ=0.3. The value of γ was chosen arbitrarily. However, it was chosen in a way that the geometry would be prevented from scaling down such that the internal surface nodes would intersect, which would result in numerical errors. The unloaded aortic configuration $\Omega (X_{aorta}^{unload},0)$ was then created (Equation (20); Figure 5) by scaling the obtained displacement field from the initial aortic geometry $\Omega (X^{MRI},0)$.
(18)$$\Omega \left(x_{int},{\overset{̿}{\sigma }}^{*}\right)=S(\Omega (X^{MRI},0),\;p_{sys})$$
(19)$$U_{aorta}^{disp}=x_{int}-X^{MRI}$$
(20)$$X_{aorta}^{unload}\;\;=X^{MRI}-\gamma \cdot U_{aorta}^{disp}$$

Diastolic and systolic configurations—$\Omega (x_{aorta,dias}^{exp})$ (Figure 2B; Figure 5) and $\Omega (x_{aorta,sys}^{exp})$ (Figure 5)—were generated in a similar way as the cylindrical geometry (Figure 5; Equations (21) and (22)) for three values of shear modulus-like mechanical parameter (c = 0.9 MPa, 1 MPa and 1.1 MPa; with same values of $p_{dias}$ and $p_{sys}$):(21)$$\Omega (x_{aorta,dias}^{exp},{\overset{̿}{\sigma }}_{dias}^{exp})=S(\Omega (X_{aorta}^{unload},0),\;p_{dias})$$
(22)$$\Omega (x_{aorta,dias}^{exp},{\overset{̿}{\sigma }}_{sys}^{exp})=S(\Omega (X_{aorta}^{unload},0),\;p_{sys})$$

Contrary to the cylindrical geometry, displacement obtained from LVOT slice cine images was applied at the $\Gamma_{root}$ boundary of the aorta (Figure 2) while pressurising the unloaded geometry to systolic pressure. The magnitude of the displacement was 8.7 mm. The displacement was not in the direction of the normal to $\Gamma_{root}$. For the aortic geometry, $R_{DBP}$ and $R_{SBP}$ (Equation (17)) were created from the nodes located on the ascending aortic inner boundary (Figure 2A).

### 2.5. Simulations and Analyses

#### 2.5.1. Assessment of Algorithm Accuracy

On completion of the optimisation, the deviation (Δc) of the estimated parameter c from the true value (reference value of c) was calculated as the absolute mean difference between the estimated value and the reference value. Accuracy (ζ) of the estimation, for each reference case, was measured by:(23)$$\zeta =100-\frac{\sum_{i=1}^{n}\Delta c_{i}}{n}\cdot 100\%$$
where, n is the number of initial guesses (in this case n = 4) and $\Delta c_{i}$ is the deviation for each estimation for a given value of reference value of c. The function tolerance for lsqnonlin was set to the default value provided by MATLAB, which is 10^−6^.

#### 2.5.2. Pre-Stresses

The pre-stresses arising in the diastolic or systolic configuration are due to the deformations experienced by the geometry with respect to the unloaded configuration. Therefore, if the unloaded configuration is accurately estimated (along with mechanical parameter c), then the pre-stresses arising in the pressurised configurations would also be accurate. Therefore, for the aortic geometry, the effect of pre-stresses was indirectly determined by comparing the estimated value of γ with the value of γ used to create the reference data. 

However, since the cylindrical unloaded geometry was directly created as opposed to the aortic unloaded geometry, γ was not pre-assigned. Therefore, the pre-stresses in the diastolic configuration after estimation were compared to the pre-stresses in the reference diastolic cylindrical geometries (${\overset{̿}{\sigma }}_{dias}^{exp}$ in Equation (14)). For comparison, the differences of L^2^–norms of the principal stresses of the reference diastolic configuration geometry ($\sigma_{cylinder,dias}^{exp}$) and the estimated diastolic configuration ($\sigma_{cylinder,dias}^{est}$) were calculated using:(24)$$\sqrt{\sum_{i=1}^{n}{\left({\left|\left(\sigma_{cylinder,dias}^{exp})-(\sigma_{cylinder,dias}^{est}\right)\right|}_{i}\right)}^{2}}$$
where, i represents the element number.

#### 2.5.3. Effect of Different Initial Conditions

The algorithm was tested on four different sets of initials guesses for c and γ required to initiate the parameter estimation process (Figure 5) for the cylindrical and aortic reference data. The four sets of initial guesses for c and γ were: (i) 0.5 MPa and 1, (ii) 0.75 MPa and 0.75, (iii) 1 MPa and 0.5 and (iv) 1.5 MPa and 0.25.

The initial values for c were chosen to check if convergence to the true value was obtained when the material initially behaved flexibly (c = 0.5 MPa) or stiffly (c = 1.5 MPa). The values of 0.75 MPa and 1 MPa were chosen to check the convergence with the middle value of the stiffness range (0.5–1.5 MPa). The initial values of γ are lying within the range as previously recommended [10]. 

To check the convergence for γ to the true value for aortic reference data (0.3; cf. Sec. aortic geometry reference data), $X_{aorta}^{unload}$ in the algorithm (Figure 3) was defined as $X_{aorta}^{unload}=X^{MRI}-\gamma \cdot U_{aorta}^{disp}$ instead of $X_{aorta}^{unload}=x_{aorta,dias}^{exp}-\gamma \cdot U_{aorta}^{disp}$, since the unloaded configuration for the aorta was created from $X^{MRI}$ (Equation (20)) and not from $x_{aorta,dias}^{exp}$. 

## 3. Results

Table 1 describes the outcome of the estimated parameter values for the cylindrical and aortic geometries. It can be observed that the mechanical parameter c was slightly overestimated for the cylinder, while underestimated for the aorta. However, independent from the initial guesses, the estimations resulted in acceptable accuracies of c for the cylindrical (98.43%, 98.50%, 98.50%) and the aortic (99.15%, 99.22%, 99.20%) geometries. The differences in stresses for cylindrical geometry between the reference data and the post-estimation of mechanical parameters ranged from 0.005 MPa to 0.007 MPa. The average values of γ for the cylindrical geometry were 1.279, 1.313 and 1.340, whereas the aortic geometry resulted in average values of 0.304, 0.304 and 0.303 for reference c values of 0.9, 1 and 1.1 MPa, respectively.

## 4. Discussion

In the current study, we developed a fixed-point two-staged iterative algorithm to estimate the mechanical parameters of the ascending aorta while considering the in vivo pre-stresses. The underlying principle of the two-staged approach is to estimate mechanical parameters in a systematic manner by using IFEA on the estimated unloaded configuration, which is vital for obtaining in vivo pre-stresses. The approach enabled the accurate estimation of mechanical parameters for a cylindrical and a realistic aortic geometry using synthetically generated reference data.

### 4.1. Methodological Differences with Respect to BI

Our method is based on previously proposed methods to determine the unloaded configurations of blood vessels [6,10], with an additional ability to estimate mechanical parameters by considering the pre-stresses [7,11]. Although the BI method has proven to be effective for the mechanical characterisation of arteries [7,11], it does not calculate the pre-stresses or the unloaded configuration directly, but rather calculates an equilibrium configuration by incrementally applying in vivo pressure to the in vivo geometry. During the process of incrementally pressurising the in vivo geometry, the deformations are discarded. The resulting effect is an in vivo geometry with pre-stresses only. Nevertheless, in order to utilise the BI method, adjustments must be made to the structural solver used for FE analysis. Specifically, when incrementally applying pressure to an arterial geometry, the initial stress field needs to be updated to account for the pre-stresses accumulated from previous increments. Simultaneously, it is necessary to discard the deformations associated with the incremental pressure applied in the current step of the analysis [6]. Our approach allows for the incorporation of in vivo pre-stresses without modifying the source code of the FE structural solver. The key difference between our method and the backward incremental approach is that our method makes explicit use of the unloaded configuration to calculate stresses in the in vivo configuration, whereas the backward incremental method solves explicitly for in vivo pre-stresses while implicitly considering the unloaded configuration [6,7]. 

### 4.2. Basic Numerical Validation

To numerically validate the utility of our method, the algorithm was first implemented on a cylindrical case. The cylindrical case was chosen in order to simplify the deformation field to expansion only in the radial direction. To achieve radial direction expansion only, our cylinder case with fixed ends was chosen to be long in order to ameliorate the effect of boundary conditions. Nevertheless, the radius of the cylinder and the pressure conditions it was subjected to were similar to a normal ascending aorta. Accuracy checks were performed on the estimated mechanical parameter c by comparing it with the true value with which the reference data were produced synthetically. The minimum accuracy of the estimated c for the cylindrical geometry was more than 98%.

Following the numerical validation of the cylindrical case, the method was further numerically validated using a realistic aortic geometry and boundary conditions. The boundary conditions for the aortic geometry were chosen in order to mimic the in vivo conditions. However, since the displacement of the aortic root cannot be obtained from the in vivo unloaded configuration to the diastolic configuration, the zero-displacement boundary condition of the aortic root was chosen while pressurising the unloaded configuration to the diastolic pressure. Therefore, the aorta did not experience any axial stretch while deforming from an unloaded configuration to the configuration at the diastolic pressure. Nevertheless, the importance of axial stretch has been acknowledged since it affects transverse deformation in the radial direction due to the Poisson’s effect as well as induced longitudinal stresses [19,20]. Besides prescribing the axial stretch directly, ventricular traction force causing the aortic root displacement could also be utilised as a boundary condition to determine the axial stretch. However, information on ventricular traction force is also not available. For that reason, the unavailability of the in vivo root displacement or ventricular traction force information from the unloaded configuration to the diastolic configuration poses as a limitation of our new method, which may affect the accuracy of the mechanical parameter c. On the contrary, the value of aortic root displacement at systolic pressure with respect to the diastolic geometry could be obtained from MRI. Therefore, the aortic root displacement was prescribed at the root level while pressurising the geometry from unloaded configuration to the systolic pressure. Applying root displacement to the directly unloaded configuration while pressurising to the systolic pressure will not cause a difference in the stresses in the vessel wall when compared to pressurising the unloaded geometry to the diastolic pressure first with fixed root, and then pressurising the diastolic configuration till the systolic pressure with root displacement. This is because the deformations at the final state (systolic pressure) are independent of the deformation path for hyperplastic materials. 

The application of zero-displacement boundary condition at the root level while pressurising to diastolic pressure and a prescribed boundary while pressurising to the systolic pressure is not physiological, since the displacement of the root would be occurring at both diastolic and systolic geometry. Additionally, the displacement field $U_{aorta}^{disp}$, used to calculate the unloaded aortic geometry (Equation (19)), was determined by applying a positive (systolic) pressure to the geometry created from MRI. It is to be noted that $U_{aorta}^{disp}$ might not be a physiological deformation field that could have existed if the blood pressure was removed from the aorta. Nonetheless, the minimum accuracy of the estimated c for the aortic geometry was more than 99% regardless of the initial guesses for c in our algorithm.

Furthermore, the inclusion of the cylindrical and aortic geometrical cases in this study was specifically intended to serve as a means of rigorously testing our algorithm. Variations in geometrical inputs of the vessels will not result in any methodological changes in the algorithm for estimating the mechanical parameters. Similarly, the utilisation of MRI images to construct the aortic geometries was also driven by the objective of evaluating our algorithm’s performance. However, for the application of the algorithm in an advanced study aimed at estimating mechanical parameters in patients, employing higher-resolution medical images (such as CT scans) could potentially enhance the accuracy of the mechanical parameters.

### 4.3. Residual Stresses

A limitation of this study was the exclusion of residual stresses while calculating the pre-stresses. We consider residual stresses as stresses present in the arterial wall in its zero-pressure intact configuration, while pre-stresses arise in the artery when pressurised from its zero-pressure configuration to the loaded configuration (in our case, the diastolic configuration from MRI). Essentially, the stress state in a loaded arterial configuration is a superposition of the residual and the pre-stresses. The exclusion of residual stresses was based on the assumption that their effect is significantly lower than that of pre-stresses [7,10,11,21].

Generally, an in vivo assessment of residual stresses is not possible without a priori knowledge of tissue mechanical properties or the opening angle of the excised vessel ring [22]. Nonetheless, a previous study has demonstrated the possibility of determining the residual stresses as well as mechanical parameter estimation [23]. However, a cylindrical vessel geometry was considered in that study while the vessel experienced homogenous deformations on releasing the residual stresses. It was also noted that the release of axial and circumferential stretches depended on the vessel geometry, thereby rendering homogenous deformations in their case. However, owing to the geometry, a complex regionally varying residual stress distribution is present on the ascending aorta, which renders numerical implementation challenging [24]. As a result, we opted to adhere to our exclusion criterion for residual stresses.

### 4.4. Fitting Procedure

In addition to realistic boundary conditions, incorporating sufficient experimental data points would enable fitting relevant anisotropic constitutive models for arteries. For example, there are studies whose authors performed in vivo parameter estimation using an anisotropic Holzapfel–Gasser–Odgen (HGO) constitutive model for ascending aorta [25] and abdominal aorta [26]. However, in these studies, the arteries were assumed to be cylindrical with a homogeneous deformation field, and the axial stretch was considered as a constant parameter that was fitted to the data. Contrary to those assumptions, the deformation field on the ascending aorta is not homogenous at a given phase during the cardiac cycle, while the axial stretches are also varying with each phase of the cardiac cycle [27,28,29,30]. Furthermore, the determination of these non-homogenous deformations requires the tracking of anatomically marked local regions on the ascending aorta, which in itself is challenging and prone to being restricted only to the bifurcations and valvular plane of the ascending aorta [27,28,29]. Due to these limitations and preliminary nature of our study, a Mooney–Rivlin type of formulation was considered as the mechanical property of the cylindrical and the aortic geometry, and the fitting was limited to only two pressure conditions. Nonetheless, the principle of our method remains the same and it can be implemented on clinical data using a complex constitutive law, given that the aforementioned limitations are addressed. Hence, it remains as a future scope of our method to address the challenges to obtain accurate imaging data in terms of local non-homogenous deformations for multiple phases during cardiac cycle and fit an anisotropic constitutive model.

## 5. Conclusions

In conclusion, the method presented in this paper demonstrates the ability to estimate the mechanical parameters of an ascending aorta with admissible accuracy (error < 2%) whilst preserving pre-stresses in the aortic wall. In the future, our approach has the potential to be applied in estimating the mechanical characteristics with anisotropic constitutive models that are more applicable to arterial tissues, given that sufficient biaxial loading data are available. By utilising our method in a case–control study, valuable insights can be gained regarding the pathological alterations in the mechanical properties of tissues.

## Figures and Tables

**Figure 1 bioengineering-10-00846-f001:**
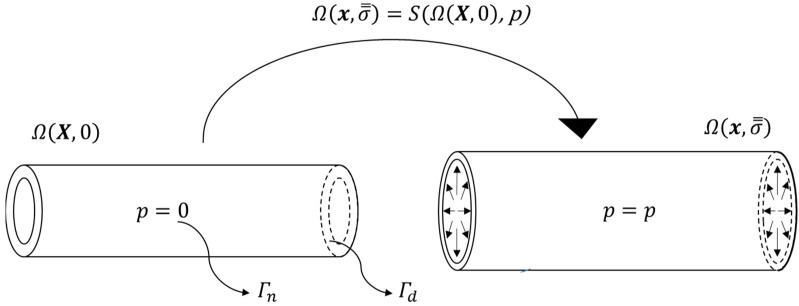
Schematic representation of computational domain, Ω, of the cylindrical geometry. The finite element system of equations are solved by a structural solver (S) by applying prescribed displacements at the opening boundaries ($\Gamma_{d})$ and pressure (p) at the inner surface ($\Gamma_{n})$. The general notation for nodes in the undeformed reference configuration and the deformed configuration are represented as X and x, respectively, with corresponding stresses as 0 and $\overset{̿}{\sigma }$.

**Figure 2 bioengineering-10-00846-f002:**
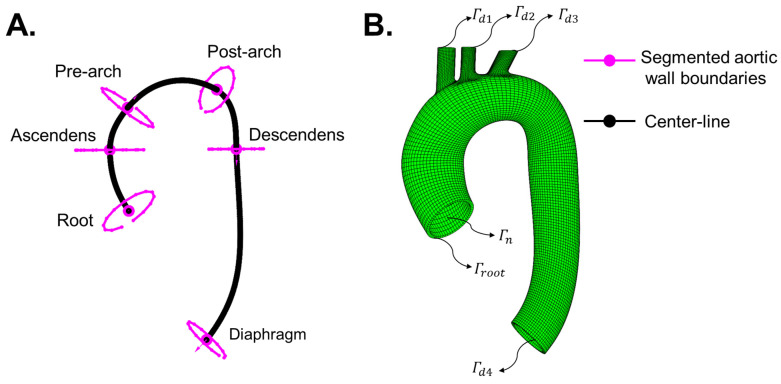
(**A**) Inner-wall aortic boundaries obtained from 3D DIXON MRI scan and centre-line connecting the lumen at six plane locations. The six plane locations were at root, ascending aorta, before brachiocephalic artery (pre-arch), after left sub-clavian artery (post-arch), descending aorta and at diaphragm levels. (**B**) Solid mesh ($\Omega (X^{MRI},0)$) of the aortic geometry, including the branches. Boundary conditions were applied on the nodes located on $\Gamma_{d1}$ to $\Gamma_{d4}$ (fixed), $\Gamma_{root}$ (fixed/displacement) and $\Gamma_{n}$ (pressure).

**Figure 3 bioengineering-10-00846-f003:**
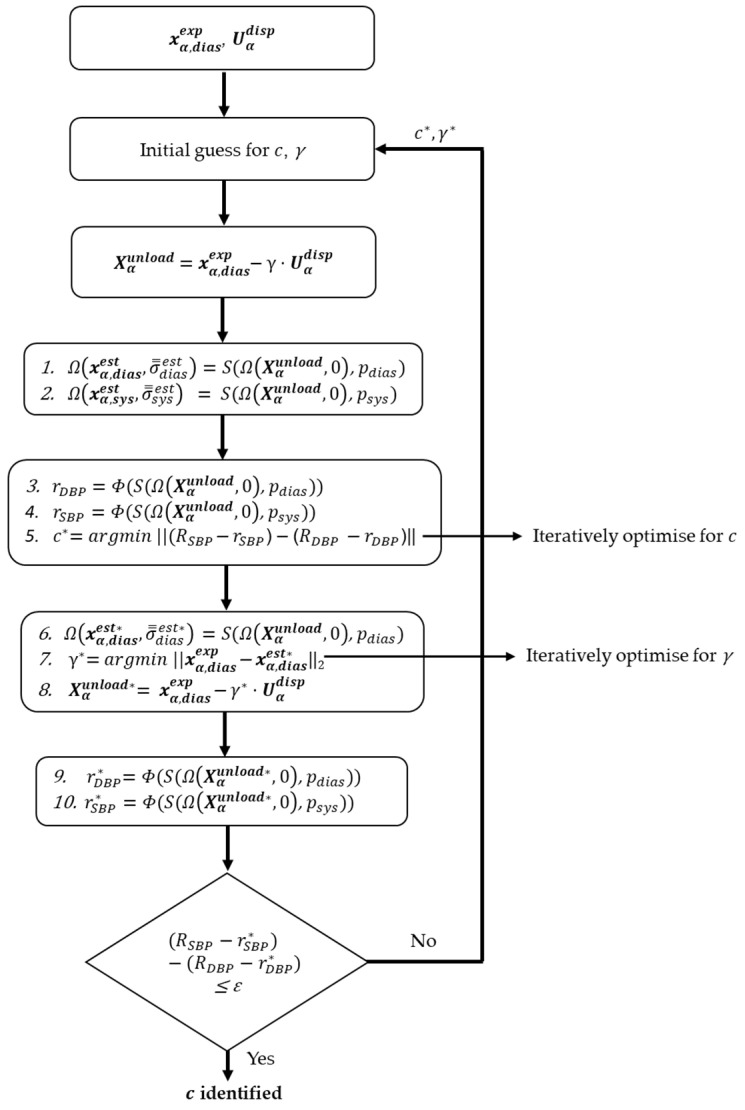
Workflow of the algorithm depicting the iterative process of estimating the mechanical parameter c and the scaling factor γ. The subscript α in $x_{\alpha ,dias}^{exp}$ and $U_{\alpha }^{disp}$, depict the type of geometry under consideration. In this paper, α = cylinder and aorta.

**Figure 4 bioengineering-10-00846-f004:**
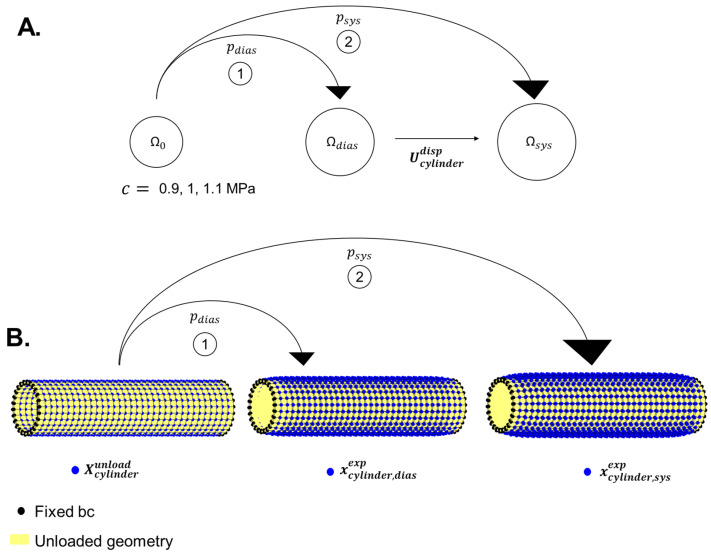
Steps involved to generate reference data for cylindrical geometry for a given unloaded configuration. (**A**) Schematic representation of the unloaded configuration $(\Omega_{0}$ = $\Omega (X_{cylinder}^{unload},0))$, diastolic configuration $(\Omega_{dias}=\Omega (x_{cylinder,dias}^{exp},{\overset{=}{\sigma }}_{dias}^{exp}))$ and systolic configuration $\Omega_{sys}=\Omega (x_{cylinder,sys}^{exp},{\overset{=}{\sigma }}_{sys}^{exp})$. (**B**) Finite element simulations depicting the nodes in unloaded ($X_{cylinder}^{unload}$), diastolic ($x_{cylinder,dias}^{exp}$) and systolic configurations ($x_{cylinder,sys}^{exp}$) associated with $\Omega_{0}$,
$\Omega_{dias}$ and $\Omega_{sys}$.

**Figure 5 bioengineering-10-00846-f005:**
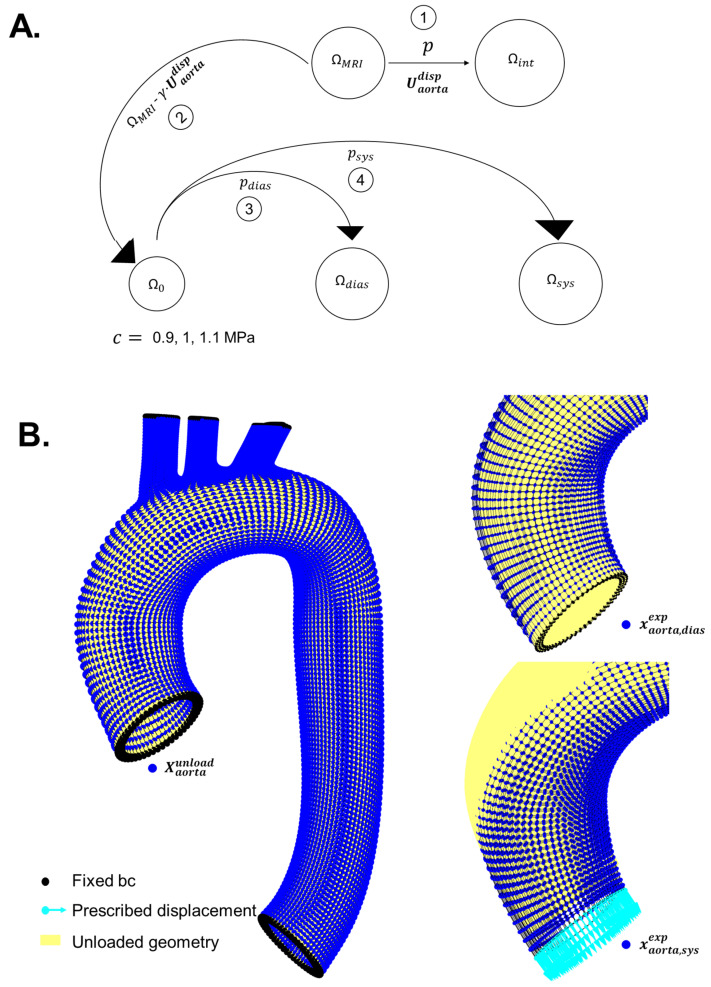
(**A**) Four-step procedure to generate reference data for aortic geometry: (i) initial MRI geometry $\Omega_{MRI}$ is inflated (till systolic pressure) to obtain an intermediate geometry $\Omega_{int}$; (ii) unloaded configuration $\Omega_{0}$ is obtained by subtracting the scaled nodal displacements from MRI geometry $\Omega_{MRI}$; (iii) subsequently pressurising $\Omega_{0}$ with diastolic and (iv) systolic pressures, resulted in $\Omega_{dias}$ and $\Omega_{sys}$. (**B**) Finite element simulations depicting the nodes in unloaded ($X_{aorta}^{unload}$), diastolic ($x_{aorta,dias}^{exp}$) and systolic configurations ($x_{aorta,sys}^{exp}$) associated with $\Omega_{0}$,
$\Omega_{dias}$ and $\Omega_{sys}$.

**Table 1 bioengineering-10-00846-t001:** Initial guesses and estimated outcomes of c and γ parameters for reference values of c for cylindrical and aortic geometries.

Geometry	Referencec (MPa)	Initial Guesses		Estimated Values		Δc (MPa)	Accuracy (ζ)
		c (MPa)	γ	c (MPa)	γ		
		0.5	1	0.930	1.260	0.030	
	0.9	0.75	0.75	0.908	1.289	0.008	98.43%
		1	0.5	0.911	1.285	0.011	
		1.5	0.25	0.914	1.281	0.014	
		0.5	1	1.032	1.293	0.032	
Cylinder	1	0.75	0.75	1.008	1.322	0.008	98.50%
		1	0.5	1.010	1.320	0.010	
		1.5	0.25	1.010	1.317	0.010	
		0.5	1	1.134	1.320	0.034	
	1.1	0.75	0.75	1.107	1.350	0.007	98.50%
		1	0.5	1.109	1.347	0.009	
		1.5	0.25	1.110	1.345	0.010	
		0.5	1	0.895	0.303	0.005	
	0.9	0.75	0.75	0.897	0.302	0.003	99.15%
		1	0.5	0.879	0.313	0.021	
		1.5	0.25	0.905	0.297	0.005	
		0.5	1	0.996	0.302	0.004	
Aorta	1	0.75	0.75	0.997	0.302	0.003	99.22%
		1	0.5	0.979	0.312	0.021	
		1.5	0.25	1.003	0.298	0.003	
		0.5	1	1.096	0.302	0.004	
	1.1	0.75	0.75	1.098	0.301	0.002	99.20%
		1	0.5	1.079	0.311	0.021	
		1.5	0.25	1.105	0.297	0.005	

## Data Availability

The authors ensure the availability of the MATLAB codes utilised in this study to any interested researcher. The request for the codes must be sent to the corresponding author.

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
