# Peer review of "Biomechanical Characterisation of Thoracic Ascending Aorta with Preserved Pre-Stresses"

_bioengineering, 2023, doi:10.3390/bioengineering10070846_

Round 1

Reviewer 1 Report

The proposed manuscript is devoted to a new identification technique for evaluation of aortic mechanical properties. The constructed algorithm is based on inverse finite element analysis. It is able to identify mechanical properties non-invasively while considering pre-stresses.

Preliminaries to the research area are provided. Recent results and methods concerning estimation of mechanical parameters and pre-stresses are carefully overviewed and described.

Details of the proposed approach are provided. In particular the authors describe the geometry creation, the finite element model, their inverse modelling algorithm taking into account existing pre-stresses as well as details regarding the creation of synthetic reference data and the performed simulations to assess the accuracy of the proposed approach.

The results of the analysis are presented showing very high accuracy of estimation of the true aortic mechanical properties.

The presentation of the main results is clear and comprehensive. From a formal point of view, all the contents seems to be correct. The results are valuable and worthy of being published taking into account their possible applications in clinical practice, in particular for understanding the pathological changes associated with the tissue.

Minor revisions are suggested to improve the quality of the exposition:

p. 1, line 156 and below, especially p. 10 line 362: The formatting of the equations should be improved moving the numbers to the right edge.

p. 5, line 194: I suggest the explanation in brackets to be moved under the equations (5).

p. 17, line 570: I suggest the conclusion to be developed by adding possible future directions of research in this field.

Reviewer 2 Report

This work proposes an inverse finite-element approach to determine the unloaded reference configuration of the aneurysmal aorta. The reverse approach is interesting but the applicability on clinical practice is not clear given the proposed imaging modalities for model development. Moreover, the literature is plenty of inverse approach for material parameters evaluation in ascending thoracic aortic aneurysm but none of them is discussed. The paper is well written but the following major comments need to be addressed:

page 2 line 67: please provide some works on ascending thoracic aortic aneurysm. there are plenty in literature for the aneurysmal aorta. Please consider to report this study

Cosentino et al. On the role of Material Properties in ascending thoracic aortic aneurysms Computers in Biology and Medicine, 2019 (109), 70-78

page 3 line 103: the normal aorta is 20 mm in diameter. why the diameter of 32 mm was selected? this not represent the aneurysmal aorta having 40 mm in diameter. the 32 mm is likely an ectasia

page 3 line 105. as before the thickness do not represent that of the aorta.

page 3 line 107. Please provide more detail on how the mesh independency was carried out.

page 5 line 182. the aorta is usually to be known as an anisotropic material. please provide explanation on the utilization of isotropic material assumption.

page 5 line 182. the aorta is usually to be known as an anisotropic material. please provide explanation on the utilization of isotropic material assumption.

page 5 line 192: the description of Cauchy stress calculation is here meaningless

page 5 line 208: to obtain two cardiac phase, dynamic CT angiography needs be performed but is not the golden-standard in aneurysm imaging. Moreover, CT is preferred over MRI. This study basically represent an application limited only to the type of imaging done on the patient and cannot therefore extended to all cases.

page 6 line 18. Here the assumption is to uniformly scale the geometry to obtain the zero-stress configuration. this is not correct given the anisotropic behavior of the vessel. Studies as theat of Martin et have demonstrated using biaxial testing and computational modeling that the model needs to be scaled differently in longitudinal versus circumferential direction

page 7 line 289. the approach here described seems to be implemented in MATLAB only. Is there any connection between MATLAB and Febio during the equation computation.

page 15 line 478: the term "validation" is misleading. the validation should be done using in-vivo or ex-vivo and thus in general using an approach that is different from the computational modeling here used. Validation is a direct comparison between the actual unloaded configuration and the estimated one. As the actual unloaded configuration is unknown, the validation is numeric only. Thus, the paragraph should be named "numerical validation".

Round 2

Reviewer 2 Report

All comments were addressed